# Integrating Spatial Configuration into Heatmap Regression Based CNNs for Landmark Localization

**Christian Payer**[1,2]                                    CHRISTIAN.PAYER@ICG.TUGRAZ.AT
[1] *Institute of Computer Graphics and Vision, Graz University of Technology, Graz, Austria*
[2] *Ludwig Boltzmann Institute for Clinical Forensic Imaging, Graz, Austria*

**Darko Štern**[2]                                          DARKO.STERN@CFI.LBG.AC.AT
**Horst Bischof**[1]                                        BISCHOF@ICG.TUGRAZ.AT
**Martin Urschler**[2,1]                                    MARTIN.URSCHLER@CFI.LBG.AC.AT

## Abstract

In many medical image analysis applications, often only a limited amount of training data is available, which makes training of convolutional neural networks (CNNs) challenging. In this work on anatomical landmark localization, we propose a CNN architecture that learns to split the localization task into two simpler sub-problems, reducing the need for large training datasets. Our fully convolutional SpatialConfiguration-Net (SCN) dedicates one component to locally accurate but ambiguous candidate predictions, while the other component improves robustness to ambiguities by incorporating the spatial configuration of landmarks. In our experimental evaluation, we show that the proposed SCN outperforms related methods in terms of landmark localization error on size-limited datasets.

**Keywords:** anatomical landmarks, localization, heatmap regression, spatial configuration

## 1. Introduction

Localization of anatomical landmarks is an important step in medical image analysis, e.g., in segmentation (Beichel et al., 2005), or registration (Johnson and Christensen, 2002). Unfortunately, locally similar structures often introduce difficulties due to ambiguity into landmark localization. To deal with these difficulties, machine learning based approaches often combine local landmark predictions with explicit handcrafted graphical models, aiming to restrict predictions to feasible spatial configurations. Thus, the landmark localization problem is simplified by separating the task into two successive steps. The first step is dedicated to locally accurate but potentially ambiguous predictions, while in the second step graphical models (Cootes et al., 1995; Felzenszwalb and Huttenlocher, 2005) eliminate ambiguities.

Recent advances in computer vision and medical imaging have mainly been driven by convolutional neural networks (CNNs) due to their superior capabilities to automatically learn important image features (LeCun et al., 2015). Unfortunately, CNNs typically need large amounts of training data. Especially in medical imaging, this requirement is hard to fulfill, due to ethical and financial concerns as well as time consuming expert annotations.

In this work, we show that the amount of required training data can be reduced with our proposed two-component SpatialConfiguration-Net (SCN), which follows the idea of handcrafted graphical models to split landmark localization into two successive steps. This

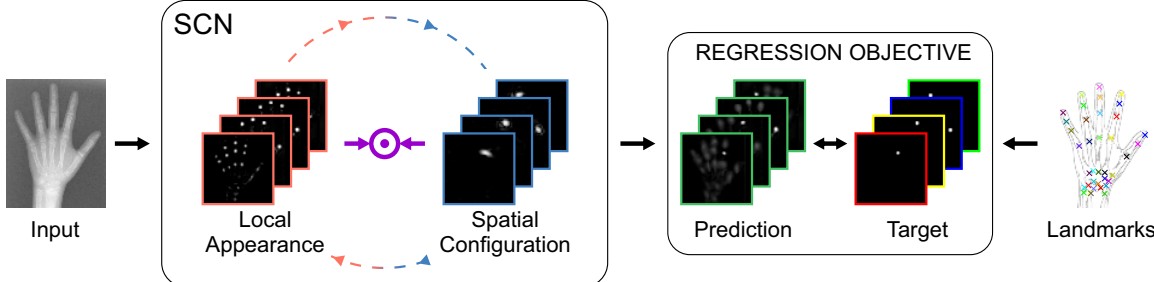

Figure 1: Landmark localization by regressing heatmaps for each landmark in our end-to-end trained fully convolutional SpatialConfiguration-Net (SCN).

extended abstract gives a short overview of the key concepts of our journal paper published in (Payer et al., 2019), while we refer the reader to the full paper for more detailed descriptions and more extensive evaluations on a variety of datasets.

## 2. Method

Our method for landmark localization is based on regressing heatmap images (Tompson et al., 2014), which encode the pseudo-probability of a landmark being located at a certain pixel position. With $N$ being the total number of landmarks, we define the target heatmap image of a landmark $L_i$, $i = \{1, ..., N\}$ as the $d$-dimensional Gaussian function $g_i(\boldsymbol{x}) : \mathbb{R}^d \to \mathbb{R}$ centered at the target landmark's groundtruth coordinate $\overset{*}{\boldsymbol{x}}_i \in \mathbb{R}^d$.

The network is set up to regress $N$ heatmaps *simultaneously* by minimizing the differences between predicted heatmaps $h_i(\boldsymbol{x})$ and the corresponding target heatmaps $g_i(\boldsymbol{x})$ in an end-to-end manner (Ronneberger et al., 2015; Shelhamer et al., 2017). In network inference, we obtain the predicted coordinate $\hat{\boldsymbol{x}}_i \in \mathbb{R}^d$ of each landmark $L_i$ by taking the coordinate, where the heatmap has its highest value.

### 2.1. SpatialConfiguration-Net

The fundamental concept of the SpatialConfiguration-Net (SCN) is the interaction between its two components (see Fig. 1). The first component takes the image as input to generate locally accurate but potentially ambiguous *local appearance* heatmaps $h_i^{\text{LA}}(\boldsymbol{x})$. Motivated by handcrafted graphical models for eliminating these potential ambiguities, the second component takes the predicted candidate heatmaps $h_i^{\text{LA}}(\boldsymbol{x})$ as input to generate inaccurate but unambiguous *spatial configuration* heatmaps $h_i^{\text{SC}}(\boldsymbol{x})$.

For $N$ landmarks, the set of predicted heatmaps $\mathbb{H} = \{h_i(\boldsymbol{x}) \mid i = 1 \ldots N\}$ is obtained by element-wise multiplication $\odot$ of the corresponding heatmap outputs $h_i^{\text{LA}}(\boldsymbol{x})$ and $h_i^{\text{SC}}(\boldsymbol{x})$ of the two components:

$$h_i(\boldsymbol{x}) = h_i^{\text{LA}}(\boldsymbol{x}) \odot h_i^{\text{SC}}(\boldsymbol{x}). \tag{1}$$

This multiplication is crucial for the SCN, as it forces both of its components to generate a response on the location of the target landmark $\overset{*}{\boldsymbol{x}}_i$, i.e., both $h_i^{\text{LA}}(\boldsymbol{x})$ and $h_i^{\text{SC}}(\boldsymbol{x})$ deliver

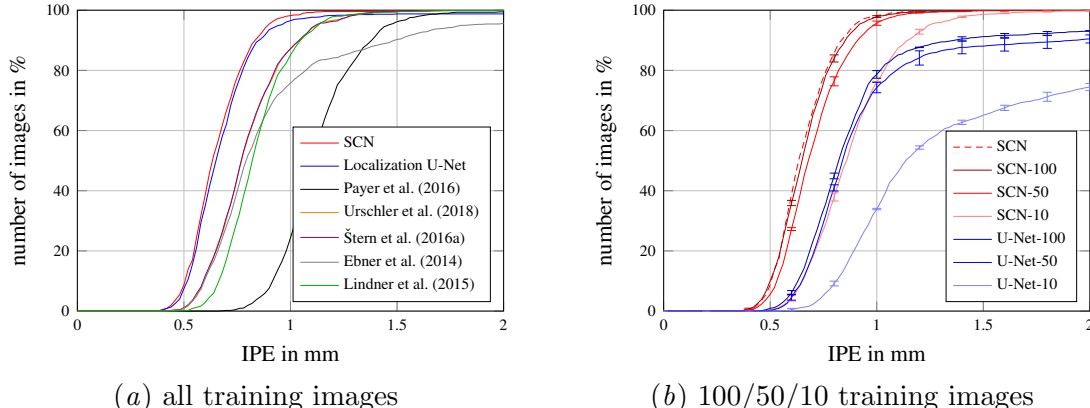

(a) all training images  (b) 100/50/10 training images

Figure 2: Cumulative distributions of the point-to-point error for 895 radiographs. (a) shows results compared with other state-of-the-art methods. (b) shows results of SCN and localization U-Net for reduced numbers of training images.

responses for $\boldsymbol{x}$ close to $\overset{*}{\boldsymbol{x}}_i$, while on all other locations one component may have a response as long as the other one does not have one.

## 3. Experiments and Results

We evaluate our proposed SCN on a dataset of 895 radiographs of left hands with 37 annotated characteristic landmarks on finger tips and bone joints. We compare our SCN to state-of-the-art random regression forests (Ebner et al., 2014; Lindner et al., 2015; Štern et al., 2016; Urschler et al., 2018), our previous CNN-based method of (Payer et al., 2016), and our implementation of a localization U-Net for heatmap regression. Results of the image-specific point-to-point errors for three-fold cross validation of the 895 radiographs are shown in Fig. 2. When using all training images, our SCN outperforms all other compared methods. Additionally, when drastically reducing the number of training images to 100, 50, and 10, respectively, our SCN greatly outperforms the localization U-Net. This confirms that splitting the localization task into predicting accurate but potentially ambiguous *local appearance* heatmaps and inaccurate but unambiguous *spatial configuration* heatmaps is especially useful when dealing with only limited amounts of training data.

## 4. Conclusion

In conclusion, we have shown how to combine information of *local appearance* and *spatial configuration* into a single end-to-end trained network for landmark localization. Our generic architecture achieves state-of-the-art results in terms of localization error, even when only limited amounts of training images are available. We are currently looking into extending our SCN regarding occluded structures and multi-object localization, and into adapting our SCN for semantic segmentation problems (see (Payer et al., 2018) for preliminary results), where structural constraints may be used in a similar manner.

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
