# OpenReview forum: "Integrating Spatial Configuration into Heatmap Regression Based CNNs for Landmark Localization"
_MIDL.io/2019/Conference/Abstract — MIDL Abstract 2019_

### Official Review · AnonReviewer1 · 2019-04-29
**Work proposes a new architecture for landmark localization using FCNs, which outperforms several compared state-of-the-art methods.**

**Rating:** 4
**Confidence:** 3

**Review:**

Summary:
Authors propose SpatialConfiguration-Net (SCN), an FCN for regressing heatmap for a given anatomical landmark. The framework consists of two networks (local accurate appearance & spatial vague location) which are fused through element-wise multiplication of the activation maps, similar to soft-attention approaches. Presented work achieves impressive results, especially when training data is scarce. Proposed model can be readily extended to 3D or other landmark localization tasks. I believe the implications of this work can be motivating for many other researchers, who can readily apply it to their work.

Strengths:
Proposed method shows impressive localization performance, especially compared to other state-of-the-art methods. Method significance becomes clearer with reducing training set size.

Weaknesses:
It is not clear what is the clinical tolerance for the landmark localization. For example, while SCN suffers a lot less with scarcer training set, is SCN-10 still relevant for the task? Definition of such a clinical tolerance can be especially handy when claiming applicability of the method, given that the quantitative results show exactly the %of test images within an error margin for compared methods.

Comments:
- Authors should consider also mentioning more straightforward quantitative metrics such as average landmark localization error.

- It is intuitive that a network only responsible of vaguely approximating location of a certain landmark would need less training samples (spatial configuration heatmaps). Hence, even with very little data, IPE can be very low for SCN; while local appearance heatmaps are far off from the ground truth. As the training set size increase, local appearance heatmaps can correct the fine localization of the landmarks. I would recommend authors to consider observing the activation map H^{LA} and H^{SC} as the training set size is reduced/increased.

---

### Official Review · AnonReviewer2 · 2019-04-30
**Interesting approach for landmark localization in medical imaging**

**Rating:** 3
**Confidence:** 2

**Review:**

This is an abstract version of a recently accepted journal paper. The paper proposes several novel concepts to include spatial configurations into object detection algorithms which can be useful for medical imaging applications. The approach alleviates the need for large datasets (which are typically difficult to acquire in medical imaging) by splitting the localization task into two simpler sub-problems.

Pros:
The proposed approach outperforms other approaches on size-limited datasets in terms of landmark localization error.
Given the fact that the work has been recently accepted in a good journal, it is likely going to be of interest to the conference audience.

Cons:
The improved performance only seems to significant if small datasets are used. Given enough data, a U-Net type localization network for heatmap regression can perform comparably.

---

### Decision · Program_Chairs · 2019-05-06
**Acceptance Decision**

Accept